# Learning to Learn Variational Semantic Memory

**Xiantong Zhen**[1,2*], **Yingjun Du**[1*], **Huan Xiong**[3,4], **Qiang Qiu**[5], **Cees G. M. Snoek**[1] , **Ling Shao**[2,4]

[1]AIM Lab, University of Amsterdam, Netherlands
[2]Inception Institute of Artificial Intelligence, Abu Dhabi, UAE
[3]Harbin Institute of Technology, Harbin, China
[4]Mohamed bin Zayed University of Artificial Intelligence, Abu Dhabi, UAE
[5]Electrical and Computer Engineering, Purdue University, USA
{x.zhen,y.du,cgmsnoek}@uva.nl

## Abstract

In this paper, we introduce *variational semantic memory* into meta-learning to acquire long-term knowledge for few-shot learning. The variational semantic memory accrues and stores semantic information for the probabilistic inference of class prototypes in a hierarchical Bayesian framework. The semantic memory is grown from scratch and gradually consolidated by absorbing information from tasks it experiences. By doing so, it is able to accumulate long-term, general knowledge that enables it to learn new concepts of objects. We formulate memory recall as the variational inference of a latent memory variable from addressed contents, which offers a principled way to adapt the knowledge to individual tasks. Our variational semantic memory, as a new long-term memory module, confers principled recall and update mechanisms that enable semantic information to be efficiently accrued and adapted for few-shot learning. Experiments demonstrate that the probabilistic modelling of prototypes achieves a more informative representation of object classes compared to deterministic vectors. The consistent new state-of-the-art performance on four benchmarks shows the benefit of variational semantic memory in boosting few-shot recognition.

## 1 Introduction

Memory plays an essential role in human intelligence, especially for aiding learning and reasoning in the present. In machine intelligence, neural memory [22, 73, 23] has been shown to enhance neural networks by augmentation with an external memory module. For instance, episodic memory storing past experiences helps reinforcement learning agents adapt more quickly and improve sample efficiency [7, 55, 24]. Memory is well-suited for few-shot learning by meta-learning in that it offers an effective mechanism to extract inductive bias [21] by accumulating prior knowledge from a set of previously observed tasks. One of the primary issues when designing a memory module is deciding what information should be memorized, which usually depends on the problems to solve. Though being highly promising, it is non-trivial to learn to store useful information in previous experience, which should be as non-redundant as possible. Existing few-shot learning works with external memory typically store the information from the support set of the current task [41, 70, 57, 40, 30], focusing on learning the access mechanism, which is assumed to be shared across tasks. The memory used in these works is short-term with limited capacity [18, 39] in that long-term information is not well retained, despite the importance for efficiently learning new tasks.

Semantic memory, also known as conceptual knowledge [43, 67, 59], refers to general facts and common world knowledge gathered throughout our lives [52]. It enables humans to quickly learn

---

new concepts by recalling the knowledge acquired in the past [59]. Compared to episodic memory, semantic memory has been less studied [67, 59], despite its pivotal role in remembering the past and imagining the future [29]. By its very nature, semantic memory can provide conceptual context to facilitate novel event construction [28] and support a variety of cognitive activities, e.g., object recognition [5]. We draw inspiration from the cognitive function of semantic memory and introduce it into meta-learning to learn to collect long-term semantic knowledge for few-shot learning.

In this paper, we propose an external memory module to accrue and store long-term semantic information gained from past experiences, which we call variational semantic memory. The function of semantic memory closely matches that of prototypes [61, 1], which identify the semantics of objects in few-shot classification. The semantic knowledge accumulated in the memory helps build the new object concepts represented by prototypes typically obtained from only one or few samples [61]. We apply our variational semantic memory module to the probabilistic inference of class prototypes modelled as distributions. The probabilistic prototypes obtained are more informative and therefore better represent categories of objects compared to deterministic vectors [61, 1]. We formulate the memory recall as a variational inference of the latent memory, which is an intermediate stochastic variable. This offers a principled way to retrieve information from the external memory and incorporate it into the inference of class prototypes for each individual task. We cast the optimization as a hierarchical variational inference problem in the Bayesian framework; the parameters of the inference of prototypes are jointly optimized in conjunction with the memory recall and update. The semantic memory is gradually consolidated throughout the course of learning by updating the knowledge from new observations in each experienced task via an attention mechanism. The long-term semantic knowledge on seen object categories is acquired, maintained and enhanced during the learning process. This contrasts with existing works [57, 70] in which the memory stores data from the support set and therefore only considers the short term. In our memory, each entry stores semantics representing a distinct object category by summarizing feature representations of class samples. This reduces redundant information and saves storage overhead. More importantly it avoids collapsing memory reading and writing into single memory slots [22, 74], which ensures that rich context information is provided for better construction of new concepts.

To summarize our three contributions: *i*) We propose variational semantic memory, a long-term memory module, which learns to acquire semantic information and enables new concepts of object categories to be quickly learned for few-shot learning. *ii*) We formulate the memory recall as a variational inference problem by introducing the latent memory variable, which offers a principled way to retrieve relevant information that fits with specific tasks. *iii*) We introduce variational semantic memory into the probabilistic inference of prototypes modelled as distributions rather than deterministic vectors, which provides more informative representations of class prototypes.

## 2 Method

Few-shot classification is commonly learned by constructing $T$ few-shot tasks from a large dataset and optimizing the model parameters on these tasks. A task, also called an episode, is defined as an $N$-way $K$-shot classification problem [70, 50]. An episode is drawn from a dataset by randomly sampling a subset of classes. Data points in an episode are partitioned into a support $S$ and query $Q$ set. We adopt the episodic optimization [70], which trains the model in an iterative way by taking one episode-update at a time. The update of the model parameters is defined by a variational learning objective, which is based on an evidence lower bound (ELBO) [6]. Different from traditional machine learning tasks, meta-learning for few-shot classification trains the model on the meta-training set, and evaluates on the meta-test set, whose classes are not seen during meta-training.

In this work, we develop our method based on the prototypical network (ProtoNet) [61]. Specifically, the prototype $\mathbf{z}_n$ of an object class $n$ is obtained by: $\mathbf{z}_n = \frac{1}{K} \sum_k \Phi(\mathbf{x}_{n,k})$ where $\Phi(\mathbf{x}_{n,k})$ is the feature embedding of the sample $\mathbf{x}_{n,k}$, which is usually obtained by a convolutional neural network. For each query sample $\mathbf{x}$, the distribution over classes is calculated based on the softmax over distances to the prototypes of all classes in the embedding space:

$$p(\mathbf{y}_n = 1|\mathbf{x}) = \frac{\exp(-d(\Phi(\mathbf{x}), \mathbf{z}_n))}{\sum_{n'} \exp(-d(\Phi(\mathbf{x}), \mathbf{z}_{n'}))}, \tag{1}$$

where $\mathbf{y}$ denotes a random one-hot vector, with $\mathbf{y}_n$ indicating its $n$-th element, and $d(\cdot, \cdot)$ is some distance function. Due to its non-parametric nature, the ProtoNet enjoys high flexibility and efficiency, achieving great success in few-shot learning.

The ideal prototypical representation should be expressive and encompass enough intra-class variance, while being distinguishable between different classes. In the literature [61, 1], however, the prototypes are commonly modeled by a single or multiple deterministic vectors obtained by average pooling of only a few samples or clustering. Hence, they are not sufficiently representative of object categories. Moreover, uncertainty is inevitable due to the scarcity of data, which should also be encoded into the prototypical representations. In this paper, we derive a probabilistic latent variable model by modeling prototypes as distributions, which are learned by variational inference.

## 2.1 Variational Prototype Inference

We introduce the probabilistic modeling of class prototypes, in which we treat the prototype $\mathbf{z}$ of each class as a distribution. In the few-shot learning scenario, to find $\mathbf{z}$ is to infer the posterior $p(\mathbf{z}|\mathbf{x}, \mathbf{y})$, where $(\mathbf{x}, \mathbf{y})$ denotes the sample from the query set $\mathcal{Q}$. We derive a variational inference framework to solve $\mathbf{z}$ by leveraging the support set $S$.

Consider the conditional log-likelihood in a probabilistic latent variable model, where we incorporate the prototype $\mathbf{z}$ as the latent variable

$$\log \Big[ \prod_{i=1}^{|Q|} p(\mathbf{y}_i|\mathbf{x}_i) \Big] = \log \Big[ \prod_{i=1}^{|Q|} \int p(\mathbf{y}_i|\mathbf{x}_i, \mathbf{z}) p(\mathbf{z}|\mathbf{x}_i) d\mathbf{z} \Big], \tag{2}$$

where $p(\mathbf{z}|\mathbf{x}_i)$ is the conditional prior in which we make the prototype dependent on $\mathbf{x}_i$. In general, it is intractable to directly solve the posterior, and usually we resort to a variational distribution to approximate the true posterior by minimizing the KL divergence:

$$D_{\mathrm{KL}}[q(\mathbf{z}|S)||p(\mathbf{z}|\mathbf{x}, \mathbf{y})], \tag{3}$$

where $q(\mathbf{z}|S)$ is the variational posterior that makes the prototype $\mathbf{z}$ dependent on the support set $S$ to leverage the meta-learning setting for few-shot classification. By applying the Baye's rule, we obtain

$$\log \Big[ \prod_{i=1}^{|Q|} p(\mathbf{y}_i|\mathbf{x}_i) \Big] \geq \sum_{i=1}^{|Q|} \Big[ \mathbb{E}_{q(\mathbf{z}|S)} \big[ \log p(\mathbf{y}_i|\mathbf{x}_i, \mathbf{z}) \big] - D_{\mathrm{KL}}(q(\mathbf{z}|S)||p(\mathbf{z}|\mathbf{x}_i)) \Big], \tag{4}$$

which is the ELBO of the conditional log-likelihood in (2). In practice, the variational posterior $q(\mathbf{z}|S)$ is implemented by a neural network that takes the average feature representations of samples in the support set $S$ and returns the mean and variance of the prototype $\mathbf{z}$. This can be directly adopted as the optimization objective for the variational inference of the prototype. While inheriting the flexibility of the prototype based few-shot learning [61, 1], our probabilistic inference enhances its class expressiveness by exploring higher-order information, i.e., variance, beyond a single or multiple deterministic mean vectors of samples in each class. More importantly, the probabilistic modeling provides a principled way of incorporating prior knowledge acquired from experienced tasks. In what follows, we introduce the external memory to augment the probabilistic latent model for enhanced variational inference of prototypes.

## 2.2 Variational Semantic Memory

We introduce the variational semantic memory to accumulate and store the semantic information from previous tasks for the inference of prototypes of new tasks. The knowledge on objects in the memory is consolidated episodically by seeing more object instances, which enables conceptual representations of new objects to be quickly built up for novel categories in tasks to come.

To be more specific, we deploy an external memory unit $M$ which stores a key-value pair in each row of the memory array as [22]. The keys are the average feature representations of images from the same classes and the values are their corresponding class labels. The semantics of object categories in the memory provide context for quickly learning concepts of new object categories by seeing only a few examples in the current tasks. In contrast to most existing external memory modules [57, 47, 8], our variational semantic memory module stores semantic information by summarizing samples from individual categories, and therefore our memory module requires relatively light storage overhead, enabling more efficient retrieval of content from the memory.

**Memory recall and inference** It is pivotal to recall relevant information from the external memory and adapt it to learning new tasks when working with neural memory modules. When recalling a memory, it is not simply a read out; the content from the memory must be processed in order to fit the data in a specific task [47, 22, 73]. We regard memory recall as a decoding process of chosen content in the memory, which we accomplish via variational inference, instead of simply reading out the raw content from the external memory and directly incorporating it into specific tasks.

To this end, we introduce an intermediate stochastic variable, referred to as the latent memory $\mathbf{m}$. We cast the retrieval of memory into the inference of $\mathbf{m}$ from the addressed memory $M$; the memory addressing is based on the similarity between the content in the memory and the support set from the current task. The latent memory $\mathbf{m}$ is inferred to connect the accrued semantic knowledge stored in the long-term memory to the current task, which is seamlessly coupled with the prototype inference under a hierarchical Bayesian framework.

From a Bayesian perspective, the prototype posterior can be inferred by marginalizing over the latent memory variable $\mathbf{m}$:

$$q(\mathbf{z}|S) = \int q(\mathbf{z}|\mathbf{m}, S)p(\mathbf{m}|S)d\mathbf{m}, \tag{5}$$

where $q(\mathbf{z}|\mathbf{m}, S)$ indicates that the prototype $\mathbf{z}$ is now dependent on the support set $S$ and the latent memory $\mathbf{m}$. To leverage the external memory $M$, we design a variational approximation $q(\mathbf{m}|M, S)$ to the posterior over the latent memory $\mathbf{m}$ by inferring from $M$ conditioned on $S$:

$$q(\mathbf{m}|M, S) = \sum_{a=1}^{|M|} p(\mathbf{m}|M_a, S)p(a|M, S). \tag{6}$$

Here, $a$ is the addressed categorical variable, $M_a$ denotes the corresponding memory content at address $a$, and $|M|$ represents the memory size, i.e., the number of memory entries.

We establish a hierarchical Bayesian framework for the variational inference of prototypes:

$$\tilde{q}(\mathbf{z}|M, S) = \sum_{a=1}^{|M|} p(a|M, S) \int q(\mathbf{z}|S, \mathbf{m})p(\mathbf{m}|M_a, S)d\mathbf{m}, \tag{7}$$

which is shown as a graphical model in Figure 1. We use the support set $S$ and memory $M$ to generate the categorical variable $a$ to address the external memory, and then fetch the content $M_a$ to infer the latent memory $\mathbf{m}$, which is incorporated as a conditional variable to assist $S$ in the inference of the prototype $\mathbf{z}$. This offers a principled way to incorporate semantic knowledge and build up the prototypes of novel object categories. It mimics the cognitive mechanism of the human brain in learning new concepts by associating them with related concepts learned in the past [29]. Moreover, it naturally handles ambiguity and uncertainty when recalling memory better than the common strategy of using a deterministic transformation [22, 73].

When $a$ is given, $\mathbf{m}$ only depends on $M_a$ and no longer relies on $S$. Therefore, we can attain $p(\mathbf{m}|M_a, S) = p(\mathbf{m}|M_a)$ by safely dropping $S$, which gives rise to:

$$q(\mathbf{m}|M, S) = \sum_{a=1}^{|M|} p(\mathbf{m}|M_a)p(a|M, S). \tag{8}$$

Since the memory size is finite, bounded by the number of seen classes, we further approximate $q(\mathbf{m}|M, S)$ empirically by

$$q(\mathbf{m}|M, S) = \sum_{a=1}^{|M|} \lambda_a p(\mathbf{m}|M_a), \ \ \lambda_a = \frac{\exp\big(g(M_a, S)\big)}{\sum_i \exp\big(g(M_i, S)\big)}, \tag{9}$$

where $M_a$ is the memory slot and stores the average feature representation of samples in each category that are seen in the learning stage, and $g(\cdot, \cdot)$ is a learnable similarity function, which we implement as a dot product for efficiency by taking the averages of samples in $M_i$ and $S$, respectively.

Thus, the prototype inference can now be approximated by Monte Carlo sampling:

$$\tilde{q}(\mathbf{z}|M, S) \approx \frac{1}{J}\sum_{j=1}^{J} q(\mathbf{z}|\mathbf{m}^{(j)}, S), \ \ \mathbf{m}^{(j)} \sim \sum_{a=1}^{|M|} \lambda_a p(\mathbf{m}|M_a), \tag{10}$$

where $J$ is the number of Monte Carlo samples.

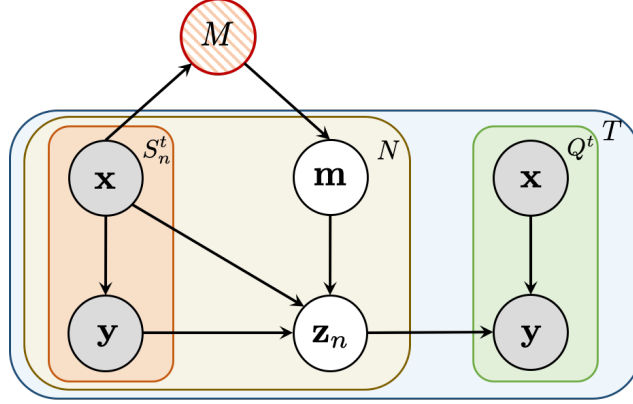

Figure 1: Graphical illustration of the proposed probabilistic prototype inference with variational semantic memory. $M$ is the semantic memory module. $S_n^t$ denotes the samples from the $n$-th class in the support set in each $t$ task. $Q^t$ is the query set. $T$ is the number of tasks, and $N$ is the number of classes in each task.

**Memory update and consolidation**    The memory update is an important operation in the maintenance of memory, which should be able to effectively absorb new useful information to enrich memory content. We draw inspiration from the concept formation process in the human cognitive function [29]: the concept of an object category is formed and grown by seeing a set of similar objects of the same category. The memory is built from scratch and gradually consolidated by being episodically updated with knowledge observed from a series of related tasks. We adopt an attention mechanism to refresh content in the memory by taking into account the structural information of data.

To be more specific, the memory is empty at the beginning of the learning. When a new task arrives, we directly append the mean feature representation of data from a given category to the memory entries if this category is not seen. Otherwise, for seen categories, we update the memory content with new observed data from the current task using self-attention [68] similar to the graph attention mechanism [69]. This enables the structural information of data to be better explored for memory update. We first construct the graph with respect to the memory $M_c$ to be updated. The nodes of the graph are a set of feature representations: $H_c = \{\mathbf{h}_c^0, \mathbf{h}_c^1, \mathbf{h}_c^2, \ldots, \mathbf{h}_c^{\mathcal{N}_c}\}$, where $\mathbf{h}_c^{\mathcal{N}_c} \in \mathbb{R}^d$, $\mathcal{N}_c = |S_c \cup Q_c|$, $\mathbf{h}_c^0 = M_c$, $\mathbf{h}_c^{i>0} = h_\phi(\mathbf{x}_c^i)$, $h_\phi(\cdot)$ is the convolutional neural network for feature extraction, and $\mathbf{x}_c^i \in \{S_c \cup Q_c\}$ contains all samples including both the support and query set from the $c$-th category in the current task.

We use the nodes $H_c$ on the graph to generate a new representation of memory $M_c$, which better explores structural information of data. To do so, we need to compute a *similarity coefficient* between $M_c$ and the nodes $\mathbf{h}_c^i$ on the graph. We implement this by a single-layer feed-forward neural network parameterized by a weight vector $\mathbf{h} \in \mathbb{R}^{2d}$, that is, $e_{M_c}^i = \mathbf{w}^\top [M_c, \mathbf{h}_c^i]$ with $[\cdot, \cdot]$ being a concatenation operation. Here, $e_{M_c}^i$ indicates the importance of node $i$'s features to node $M_c$. In practice, we use the following normalized similarity coefficients [69]:

$$\beta_{M_c}^i = \text{softmax}_i(e_{M_c}^i) = \frac{\exp(\text{LeakyReLU}\left(\mathbf{w}^\top [M_c, \mathbf{h}_c^i]\right))}{\sum_{j=0}^{\mathcal{N}_c} \exp(\text{LeakyReLU}\left(\mathbf{w}^\top [M_c, \mathbf{h}_c^j]\right))}. \tag{11}$$

We can now compute a linear combination of the feature representations of the nodes on the graph as the final output representation of $\bar{M}_c$:

$$\bar{M}_c = \sigma \left( \sum_{i=0}^{\mathcal{N}_c} \beta_{M_c}^i \mathbf{h}_c^i \right), \tag{12}$$

where $\sigma(\cdot)$ is a nonlinear activation function, e.g., softmax. The graph attention operation can effectively find and assimilate the most useful information from the samples in the new task. We update the memory content with an attenuated weighted average,

$$M_c \leftarrow \alpha M_c + (1 - \alpha) \bar{M}_c, \tag{13}$$

where $\alpha \in (0, 1)$ is a hyperparameter. This operation allows useful information to be retained in the memory, while erasing less relevant or trivial information.

## 2.3 Objective

To train the model, we adopt stochastic gradient variational Bayes [31] and implement it using deep neural networks for end-to-end learning. By combining (4), (9) and (7), we obtain the following objective for the hierarchical variational inference:

$$\underset{\{\phi,\theta,\psi,\varphi\}}{\arg\min} - \sum_t^T \left[ \sum_n^N \left[ \sum_{(\mathbf{x}_i^t,\mathbf{y}_i^t)\in Q_n^t}^{|Q_n^t|} \left[ \sum_\ell^L \left[ \log p(\mathbf{y}_i^t|h_\phi(\mathbf{x}_i^t),\mathbf{z}_n^{(\ell)}) \right] + \right. \right. \right.$$

$$D_{\mathrm{KL}}(\frac{1}{J}\sum_{j=1}^J \tilde{q}_\varphi(\mathbf{z}_n|\mathbf{m}^{(j)},\bar{\mathbf{h}}_{S_n^t})||p_\theta(\mathbf{z}_n|h_\phi(\mathbf{x}_i^t))] + D_{\mathrm{KL}}(\sum_a^{|M|}\lambda_a p_\psi(\mathbf{m}|M_a)||p_\psi(\mathbf{m}|\bar{\mathbf{h}}_{S_n^t}))] \right],$$
$$(14)$$

where $\mathbf{z}^{(\ell)} \sim \frac{1}{J}\sum_{j=1}^J q_\varphi(\mathbf{z}|\mathbf{m}^{(j)},S_n^t)$, $\mathbf{m}^{(j)} \sim \sum_{a=1}^{|M|}\lambda_a p_\psi(\mathbf{m}|M_a)$, $L$ and $J$ are numbers of Monte Carlo samples, $\bar{\mathbf{h}}_{S_n^t} = \frac{1}{|S_n^t|}\sum_{\mathbf{x}\in S_n^t} h_\phi(\mathbf{x})$, and $n$ denotes the $n$-th class. To enable back propagation, we adopt the reparameterization trick [31] for sampling $\mathbf{z}$ and $\mathbf{m}$. The third term in (14) essentially serves to constrain the inferred latent memory to ensure that it is relevant to the current task. Here, we make the parameters shared by the prior and the posterior for $\mathbf{m}$, and we also amortize the inference of prototypes across classes [20], which involves using the samples $S_n^t$ from each class to infer their prototypes individually. In practice, the log-likelihood term is implemented as a cross entropy loss between predictions and ground-truth labels. The conditional probabilistic distributions are set to be diagonal Gaussian. We implement them using multi-layer perceptrons with the amortization technique and the reparameterization trick [31, 54], which take the conditionals as input and output the parameters of the Gaussian. In addition, we implement the model with the objective in (4), which we refer to as the variational prototypical network.

## 3 Related Work

**Meta-Learning**   Meta-learning, *or learning to learn* [60, 64], for few-shot learning [32, 50, 57, 15, 70, 61] addresses the fundamental challenge of generalizing across tasks with limited labelled data. Meta-learning approaches for few-shot learning differ in the way they acquire inductive biases and adopt them for individual tasks [21]. They can be roughly categorized into four groups. Those in the first group are based on distance metrics and generally learn a shared/adaptive embedding space in which query images can be accurately matched to support images for classification [70, 61, 58, 42, 78, 1, 9]. Those based on optimization try to learn an optimization algorithm that is shared across tasks, and can be adapted to new tasks, enabling learning to be conducted efficiently and effectively [50, 2, 15, 16, 21, 66, 56, 77, 49]. The third group explicitly learns a base-learner that incorporates knowledge acquired by the meta-learner and effectively solves individual tasks [20, 4, 81]. In the fourth group, a memory mechanism has been incorporated. Usually, an external memory module is deployed to rapidly assimilate new data of unseen tasks, which is used for quick adaptation or to make decisions [57, 40, 41, 38]. The methods from different groups are not necessarily exclusive, and they can be combined to improve performance [66]. In addition, meta-learning has also been explored for reinforcement learning [37, 7, 72, 12, 14, 55] and other tasks [3, 76].

**Prototypes**   The prototypical network is one of most successful meta-learning models for few-shot learning [70, 48, 25, 17]. It learns to project samples into a metric space in which classification is conducted by computing the distance from query samples to class prototypes. Allen et al. [1] introduced an infinite mixture of prototypes that represents each category of objects by multiple clusters. The number of clusters is inferred from data by non-parametric Bayesian methods [48, 25]. Recently, Triantafillou et al. [66] combined the complementary strengths of prototypical networks and MAML [15] by leveraging their respective effective inductive bias and flexible adaptation mechanism for few-shot learning. Our work improves the prototypical network by probabilistic modeling of prototypes, inheriting the effectiveness and flexibility of the ProtoNet and further enriching the expressiveness of prototypes by the external memory mechanism.

**Memory**   It has been shown that neural networks with memory, such as the long-short term memory [26] model, are capable of meta-learning [27]. Recent works augment neural networks with an external memory module to improve their learning capability [57, 44, 73, 23, 30, 40, 41, 47, 74, 75]. In few-shot learning, existing work with external memory mainly store the information contained in

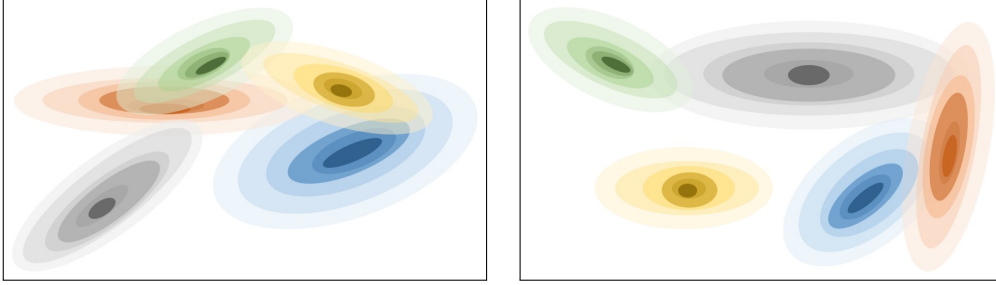

Figure 2: Prototype distributions of our variational prototype network without (left) and with variational semantic memory (right), where different colors indicate different categories. With the memory the prototypes become more distinctive and distant from each other, with less overlap.

Table 1: Benefit of variational prototype network over ProtoNet [61] in (%) on *mini*ImageNet, *tiered*ImageNet and CIFAR-FS.

|  | *mini*ImageNet, 5-way | | *tiered*ImageNet, 5-way | | CIFAR-FS, 5-way | |
| --- | --- | --- | --- | --- | --- | --- |
|  | 1-shot | 5-shot | 1-shot | 5-shot | 1-shot | 5-shot |
| ProtoNet | $47.40 \pm 0.60$ | $65.41 \pm 0.52$ | $53.31 \pm 0.89$ | $72.69 \pm 0.74$ | $55.50 \pm 0.70$ | $72.01 \pm 0.60$ |
| **Variational prototype network** | $52.11 \pm 1.70$ | $66.13 \pm 0.83$ | $55.13 \pm 1.88$ | $73.71 \pm 0.84$ | $61.35 \pm 1.60$ | $75.72 \pm 0.90$ |

Table 2: Benefit of variational semantic memory in our variational prototype network over alternative memory modules [22, 73] in (%) on *mini*ImageNet, *tiered*ImageNet and CIFAR-FS.

|  | *mini*ImageNet, 5-way | | *tiered*ImageNet, 5-way | | CIFAR-FS, 5-way | |
| --- | --- | --- | --- | --- | --- | --- |
|  | 1-shot | 5-shot | 1-shot | 5-shot | 1-shot | 5-shot |
| Variational prototype network | $52.11 \pm 1.70$ | $66.13 \pm 0.83$ | $55.13 \pm 1.88$ | $73.71 \pm 0.84$ | $61.35 \pm 1.60$ | $75.72 \pm 0.90$ |
| w/ Rote Memory | $53.15 \pm 1.81$ | $66.92 \pm 0.78$ | $55.98 \pm 1.73$ | $74.12 \pm 0.88$ | $62.71 \pm 1.71$ | $76.17 \pm 0.81$ |
| w/ Transformed Memory | $53.85 \pm 1.71$ | $67.23 \pm 0.89$ | $56.15 \pm 1.70$ | $74.33 \pm 0.80$ | $62.97 \pm 1.88$ | $76.97 \pm 0.77$ |
| **w/ Variational semantic memory** | $\mathbf{54.73} \pm 1.60$ | $\mathbf{68.01} \pm 0.90$ | $\mathbf{56.88} \pm 1.71$ | $\mathbf{74.65} \pm 0.81$ | $\mathbf{63.42} \pm 1.90$ | $\mathbf{77.93} \pm 0.80$ |

the support set of the current task [41, 70, 40], focusing on learning the access mechanism shared across tasks. In these works, the external memory is wiped from episode to episode [18, 39]. Hence, it fails to maintain long-term information that has been shown to be crucial for efficiently learning new tasks [47, 18]. Memory has also been incorporated into generative models [8, 36, 74] and sequence modeling [34] by conditioning on the context information provided in the external memory. To store minimal amounts of data, Ramalho and Garnelo proposed a surprise-based memory module, which deploys a memory controller to select minimal samples to write into the memory [47]. In contrast to [8], our variational semantic memory adopts deterministic soft addressing, which enables us to leverage the full context of memory content by picking up multiple entries instead of a single one [8]. Our variational semantic memory is able to accrue long-term knowledge that provides rich context information for quickly learning novel tasks. Rather than directly using specific raw content or deploying a deterministic transformation [22, 73], we introduce the latent memory as an intermediate stochastic variable to be inferred from the addressed content in the memory. This enables the most relevant information to be retrieved from the memory and adapted to the data in specific tasks.

## 4   Experiments

**Datasets and settings**   We evaluate our model on four standard few-shot classification tasks: *mini*ImageNet [71], *tiered*ImageNet [53], CIFAR-FS [4] and *Omniglot* [33]. For fair comparison with previous works, we experiment with both shallow convolutional neural networks with the same architecture as in [20] and a deep ResNet-12 [42, 35, 38, 51] architecture for feature extraction. More implementation details, including optimization settings and network architectures, are given in the supplementary material. We also provide a state-of-the-art comparison on the *Omniglot* dataset and more results on the performance with other deep architectures, e.g., WRN-28-10 [79].

**Benefit of variational prototype network**   We compare against the ProtoNet [61] as our baseline model in which the prototypes are obtained by averaging the feature representations of each class. These results are obtained with shallow networks. As shown in Table 1, the proposed variational prototype network consistently outperforms the ProtoNet demonstrating the benefit brought by

Table 3: Advantage of memory update with attention mechanism.

| | *mini*ImageNet, 5-way | | *tiered*ImageNet, 5-way | | CIFAR-FS, 5-way | |
| | 1-shot | 5-shot | 1-shot | 5-shot | 1-shot | 5-shot |
|---|---|---|---|---|---|---|
| w/o Attention | $53.97 \pm 1.80$ | $67.13 \pm 0.76$ | $56.05 \pm 1.73$ | $74.27 \pm 0.85$ | $62.93 \pm 1.76$ | $76.79 \pm 0.80$ |
| **w/ Attention** | $\mathbf{54.73} \pm 1.60$ | $\mathbf{68.01} \pm 0.90$ | $\mathbf{56.88} \pm 1.71$ | $\mathbf{74.65} \pm 0.81$ | $\mathbf{63.42} \pm 1.90$ | $\mathbf{77.93} \pm 0.80$ |

Table 4: Comparison with other memory models.

| | *mini*ImageNet, 5-way | | *tiered*ImageNet, 5-way | | CIFAR-FS, 5-way | |
| | 1-shot | 5-shot | 1-shot | 5-shot | 1-shot | 5-shot |
|---|---|---|---|---|---|---|
| MANN [57] | $41.38 \pm 1.70$ | $61.73 \pm 0.80$ | $44.27 \pm 1.69$ | $67.15 \pm 0.70$ | $54.31 \pm 1.91$ | $67.98 \pm 0.80$ |
| KM [74] | $53.84 \pm 1.70$ | $67.35 \pm 0.80$ | $55.73 \pm 1.65$ | $73.36 \pm 0.70$ | $62.58 \pm 1.80$ | $77.11 \pm 0.80$ |
| **Variational semantic memory** | $\mathbf{54.73} \pm 1.60$ | $\mathbf{68.01} \pm 0.90$ | $\mathbf{56.88} \pm 1.71$ | $\mathbf{74.65} \pm 0.81$ | $\mathbf{63.42} \pm 1.90$ | $\mathbf{77.93} \pm 0.80$ |

probabilistic modeling. The probabilistic prototypes provide more informative representations of classes, which are able to encompass large intra-class variations and therefore improve performance.

**Benefit of variational semantic memory** We compare with two alternative methods of memory recall: rote memory and transformed memory [22, 73] (The implementation details of are provided in the supplementary material). As shown in Table 2, our variational semantic memory surpasses alternatives on all three benchmarks. The advantage over rote memory indicates the benefit of introducing the intermediate latent memory variable; the advantage over transformed memory demonstrates the benefit of formulating the memory recall as the variational inference of the latent memory, which is treated as a stochastic variable. To understand the empirical benefit, we visualize the distributions of prototypes obtained with/without variational semantic memory in Figure 2 on *mini*ImageNet. The variational semantic memory enables the prototypes of different classes to be more distinctive and distant from each other, with less overlap, which enables larger intra-class variations to be encompassed, resulting in improved performance.

**Benefit of attentional memory update** We investigate the benefit of the attention mechanism for memory update. Specifically, we replace the attention-based update with a mean-based one; that is, we use $\bar{M} = \frac{1}{\mathcal{N}_c} \sum_i h(\mathbf{x}_c^i)$. The experimental results are reported in Table 3. We can see that the memory update with attention mechanism performs consistently better than that using the mean-based update. This is because that the with the attention mechanism, we are able to better absorb more informative knowledge from the data of new tasks by exploring the structural information.

**Effect of memory size** We conduct this experiment on *mini*ImageNet. From Figure 3, we can see that the performance increases along with the increase in memory size. This is reasonable since larger memory provides more context information for building better prototypes. Moreover, we observe that the memory module plays a more significant role in the 1-shot setting. In this case, the prototype inferred from only one example might be insufficiently representative of the object class. Leveraging context information provided by the memory, however, compensates for the limited number of examples.

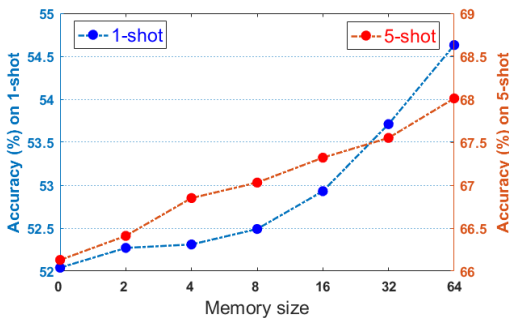

Figure 3: Effect of memory size.

**Comparison with other memory models** To demonstrate the effectiveness of our variational memory mechanism, we compare with two other representative memory models, i.e., the memory augmented neural network (MANN) [57] and the Kanerva machine (KM) [74]. MANN adopts an architecture with augmented memory capacities similar to neural Turing machines [22] while the KM deploys Kanerva's sparse distributed memory mechanism and introduces learnable addresses and reparameterized latent variables. The KM was originally proposed for generative models, but we adopt its reading and writing mechanism to our semantic memory in the meta-learning setting for few-shot classification. The results are shown in Table 4. Our variational semantic memory consistently outperforms MANN and the KM on all three datasets.

Table 5: Comparison (%) on *mini*ImageNet, *tiered*ImageNet and CIFAR-FS using a shallow feature extractor.

| | *mini*ImageNet, 5-way | | *tiered*ImageNet, 5-way | | CIFAR-FS, 5-way | |
| --- | --- | --- | --- | --- | --- | --- |
| | 1-shot | 5-shot | 1-shot | 5-shot | 1-shot | 5-shot |
| Matching Net [70] | $43.56 \pm 0.84$ | $55.31 \pm 0.73$ | - | - | - | - |
| MAML [15] | $48.70 \pm 1.84$ | $63.11 \pm 0.92$ | $51.67 \pm 1.81$ | $70.30 \pm 1.75$ | $58.90 \pm 1.91$ | $71.52 \pm 1.10$ |
| Relation Net [63] | $50.44 \pm 0.82$ | $65.32 \pm 0.70$ | $54.48 \pm 0.93$ | $65.32 \pm 0.70$ | $55.00 \pm 1.01$ | $69.30 \pm 0.80$ |
| SNAIL (32C) by [4] | $45.10 \pm 0.85$ | $55.20 \pm 0.80$ | - | - | - | - |
| GNN [19] | $50.31 \pm 0.83$ | $66.42 \pm 0.90$ | - | - | $61.90 \pm 1.03$ | $75.30 \pm 0.91$ |
| PLATIPUS [16] | $50.10 \pm 1.90$ | - | - | - | - | - |
| VERSA [20] | $53.31 \pm 1.80$ | $67.30 \pm 0.91$ | - | - | $62.51 \pm 1.70$ | $75.11 \pm 0.91$ |
| R2-D2 (64C) [4] | $49.50 \pm 0.20$ | $65.40 \pm 0.20$ | - | - | $62.30 \pm 0.20$ | $77.40 \pm 0.20$ |
| R2-D2 [11] | $51.70 \pm 1.80$ | $63.31 \pm 0.91$ | - | - | $60.20 \pm 1.80$ | $70.91 \pm 0.91$ |
| CAVIA [82] | $51.80 \pm 0.70$ | $65.61 \pm 0.60$ | - | - | - | - |
| iMAML [46] | $49.30 \pm 1.90$ | - | - | - | - | - |
| **VSM** (This paper) | $\mathbf{54.73} \pm 1.60$ | $\mathbf{68.01} \pm 0.90$ | $\mathbf{56.88} \pm 1.71$ | $\mathbf{74.65} \pm 0.81$ | $\mathbf{63.42} \pm 1.90$ | $\mathbf{77.93} \pm 0.80$ |

Table 6: Comparison (%) on *mini*ImageNet and *tiered*ImageNet using a deep feature extractor.

| | *mini*ImageNet, 5-way | | *tiered*ImageNet, 5-way | |
| --- | --- | --- | --- | --- |
| | 1-shot | 5-shot | 1-shot | 5-shot |
| SNAIL [38] | $55.71 \pm 0.99$ | $68.88 \pm 0.92$ | - | - |
| AdaResNet [41] | $56.88 \pm 0.62$ | $71.94 \pm 0.57$ | - | - |
| TADAM [42] | $58.50 \pm 0.30$ | $76.70 \pm 0.30$ | - | - |
| Shot-Free [51] | $59.04 \pm \text{n/a}$ | $77.64 \pm \text{n/a}$ | $63.52 \pm \text{n/a}$ | $82.59 \pm \text{n/a}$ |
| TEWAM [45] | $60.07 \pm \text{n/a}$ | $75.90 \pm \text{n/a}$ | - | - |
| MTL [62] | $61.20 \pm 1.80$ | $75.50 \pm 0.80$ | - | - |
| Variational FSL [80] | $61.23 \pm 0.26$ | $77.69 \pm 0.17$ | - | - |
| MetaOptNet [35] | $62.64 \pm 0.61$ | $78.63 \pm 0.46$ | $65.99 \pm 0.72$ | $81.56 \pm 0.53$ |
| Diversity w/ Cooperation [13] | $59.48 \pm 0.65$ | $75.62 \pm 0.48$ | - | - |
| Meta-Baseline [10] | $63.17 \pm 0.23$ | $79.26 \pm 0.17$ | - | - |
| Tian et al. [65] | $64.82 \pm 0.60$ | $82.14 \pm 0.43$ | $71.52 \pm 0.69$ | $86.03 \pm 0.49$ |
| **VSM** (This paper) | $\mathbf{65.72} \pm 0.57$ | $\mathbf{82.73} \pm 0.51$ | $\mathbf{72.01} \pm 0.71$ | $\mathbf{86.77} \pm 0.44$ |

**State-of-the-art comparison** As shown in Tables 5 and 6, our variational semantic memory (VSM) sets a new state-of-the-art on all few-shot learning benchmarks. On *mini*ImageNet, our model using either a shallow or deep network achieves high recognition accuracy, surpassing the second best method, i.e., VERSA [20], by a margin of $1.43\%$ on the 5-way 1-shot using a shallow network. On *tiered*ImageNet, our model again outperforms previous methods using shallow networks, e.g., MAML [15] and Relation Net [63], and deep networks, e.g., [65]. On CIFAR-FS, our model delivers $63.42\%$ on the 5-way 1-shot setting, surpassing the second best R2D2 [4] by $1.12\%$. The consistent state-of-the-art results on all benchmarks using either shallow or deep feature extraction networks validate the effectiveness of our model for few-shot learning.

## 5 Conclusion

In this paper, we introduce a new long-term memory module, named *variational semantic memory*, into meta-learning for few-shot learning. We apply it as an external memory for the probabilistic modelling of prototypes in a hierarchical Bayesian framework. The memory episodically learns to accrue and store semantic information by experiencing a set of related tasks, which provides semantic context that enables new object concepts to be quickly learned in individual tasks. The memory recall is formulated as the variational inference of a latent memory variable from the addressed content in the external memory. The memory is established from scratch and gradually consolidated by updating with knowledge absorbed from data in each task using an attention mechanism. Extensive experiments on four benchmarks demonstrate the effectiveness of variational semantic memory in learning to accumulate long-term knowledge. Our model achieves new state-of-the-art performance on four benchmark datasets, consistently surpassing previous methods. More importantly, the findings in this work demonstrate the benefit of semantic knowledge accrued through long-term memory in effectively learning novel concepts of object categories, and therefore highlight the pivotal role of semantic memory in few-shot recognition.

## Broader Impact

This work introduces the concept of semantic memory from cognitive science into the machine learning field. We use it to augment a probabilistic model for few-shot learning. The developed variational framework offers a principled way to achieve memory recall, which could also be applied to other learning scenarios, e.g., continual learning. The empirical findings indicate the potential role of neural semantic memory as a long-term memory module in enhancing machine learning models. Finally, this work will not cause any foreseeable ethical issue or societal consequence.

**Acknowledgements** The authors would like to thank all anonymous reviewers for their constructive feedback and valuable suggestions. Thanks to the partial support from national natural science foundation of China (Grants: 61976060, 61871016).

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
