[Supplementary Material]

# Supplementary Material for
# Learning to Learn Variational Semantic Memory

**Xiantong Zhen, Yingjun Du, Huan Xiong, Qiang Qiu, Cees G. M. Snoek, Ling Shao**

## A   More Results

We provide more experimental results on the *Omniglot* dataset, under the 20-way settings on *mini*ImageNet and *tiered*ImageNet datasets, comparisons with other deep architectures, e.g., WRN-10-28 [22], and further ablation studies on effect of the hyperparameter ($\alpha$) in the memory update and the effect of the Gumbel-softmax for approximating the addressing vectors in the memory recall.

On the *Omniglot* dataset, as shown in Table A1 our model consistently achieves high performance, exceeding other competitive methods (98.3% - up 0.07%) under the 20-way 1-shot setting. The results are consistent with the findings on the other datasets.

The performance under the 20-way 1-shot setting on the *mini*ImageNet and *tiered*ImageNet datasets are reported in Table A2. We compare against state-of-the-art methods, i.e., [9] and Baseline++ [3] under this setting. The proposed variational semantic memory consistently outperforms these two methods on *mini*ImageNet. We do not observe previous results under the 20-way 1-shot setting on *tiered*ImageNet and we provide our results for future comparison.

We have also experimented with one more deep architecture, i.e., WRN-28-10 [22], to compare with previous methods using the same architecture. The results are reported in Table A3. Again, our model outperforms those methods, which is consistent with the results using other architectures.

In addition, we test the impact of $\alpha$ in (13) the memory update. The value of $\alpha$ control how much information in the memory will be kept during the update with new information. The experimental results on the *mini*ImageNet dataset under both 1-shot and 5-shot setting are shown in Table A4. We can see that the performance achieves the best when the values of $\alpha$ are $0.7$ and $0.8$. This means that in each update we need to keep the majority of old information in the memory.

Finally, in the memory recall, we use softmax to generate the addressing vector, and compare it against the Gumbel-softmax approximation. The results on three datasets are shown in Table A5. We can see that Gumbel-softmax achieves comparable performance with the regular softmax. It is worth mentioning that the memory size needs to be pre-fixed when using the Gumbel-softmax approximation, while the regular softmax can deal with dynamic memories with growing size.

## B   Datasets

***mini*ImageNet**. The *mini*ImageNet is originally proposed in [20], has been widely used for evaluating few-shot learning algorithms. It consists of $60,000$ color images from 100 classes with 600 examples per class. The images have dimensions of $84 \times 84$ pixels. We follow the train/val/ test split introduced in [15], which uses 64 classes for meta-training, 16 classes for meta-validation, and the remaining 20 classes for meta-testing.

***tiered*ImageNet**. The *tiered*ImageNet dataset [16] is a larger subset of ImageNet with 608 classes ($779,165$ images) grouped into 34 higher-level categories in the ImageNet human-curated hierarchy. These categories are further divided into 20 training categories (351 classes), 6 validation categories (97 classes), and 8 testing categories (160 classes). This construction near the root of the ImageNet

hierarchy results in a more challenging, yet realistic regime with test classes that are distinctive enough from training classes.

**CIFAR-FS**. The CIFAR-FS is proposed in [2], which is randomly sampled from CIFAR-100 by using the same standard with which *mini*ImageNet has been generated. The original resolution of $32 \times 32$ pixels makes the task harder.

**Omniglot**. The Omniglot [12] is a few-shot learning dataset consisting of 1623 handwritten characters (each with 20 instances) derived from 50 alphabets. We follow a pre-processing and training procedure defined in [19]. We first resize images to $28 \times 28$ and then character classes are augmented with rotations of 90 degrees. The training, validation and test sets consist of a random split of 1100, 100, and 423 characters, respectively.

## C   Implementation details

We provide more implementation details. For the feature extraction networks, we do not use any fully connected layer after the convolutional layers. All of our models were trained via SGD with the Adam [10] optimizer. For the 5-way 5-shot model, we train using the setting of 8 tasks per batch for $100,000$ iterations and use a constant learning rate of $0.0001$. For the 5-way 1-shot model, we train with the setting of 8 tasks per batch for $150,000$ iterations and use a constant learning rate of $0.00025$. No regularization was used other than batch normalization. In the Monte Carlo sampling, we set the number $J = 150$ for $\mathbf{m}$ and to $L = 100$ for $\mathbf{z}$, which are chosen by using the validation set. The architectures of inference networks $q_\varphi(\cdot)$, $p_\psi(\cdot)$, the prior network $p_\theta(\cdot)$ and feature extraction networks $h_\phi(\cdot)$ are provided in Tables A6 and A7. The sketch of the implementation is shown in Figure A1. We implemented all models in the Tensorflow framework and tested on an NVIDIA Tesla V100.

In the experiments of comparison with alternative methods of memory recall, For rote memory, we put concatenation of the addressed memory contents with mean feature representation of the support set as input to the inference network: $q(\mathbf{z}|\bar{M}, \overline{h_\phi(\mathbf{x}_{S_n^t})})$, where $\bar{M} = \sum_a^{|\bar{M}|} \lambda_a M_a$, and $\lambda_a = \frac{g(M_a, S)}{\sum_i g(M_i, S)}$; for transformed memory, we follow the strategy in [8, 21] and pass the addressed memory $\bar{M}$ through a parameterized transformation before feeding into the inference network: $q(\mathbf{z}|T(\bar{M}), \overline{h_\phi(\mathbf{x}_{S_n^t})})$, where $T(\cdot)$ is the transformation implemented as a multi-layer perception (MLP).

Table A1: Comparison (%) on *Omniglot* using a shallow feature extractor.

|                      | *Omniglot*, **5-way** | | *Omniglot*, **20-way** | |
|----------------------|--------|--------|--------|--------|
|                      | 1-shot | 5-shot | 1-shot | 5-shot |
| Siamese Net [11]     | 96.7   | 98.4   | 88.0   | 96.5   |
| Matching net [19]    | 98.1   | 98.9   | 93.8   | 98.5   |
| MAML [5]             | 98.7 $\pm$ 0.4 | **99.9** $\pm$ 0.1 | 95.8 $\pm$ 0.3 | 98.9 $\pm$ 0.2 |
| SNAIL [13]           | 99.1 $\pm$ 0.2 | 99.8 $\pm$ 0.1 | 97.6 $\pm$ 0.3 | **99.4** $\pm$ 0.2 |
| GNN [6]              | 99.2   | 99.7   | 97.4   | 99.0   |
| VERSA [7]            | 99.7 $\pm$ 0.2 | 99.8 $\pm$ 0.1 | 97.7 $\pm$ 0.3 | 98.8 $\pm$ 0.2 |
| R2-D2 [2]            | 98.6   | 99.7   | 94.7   | 98.9   |
| IMP [1]              | 98.4 $\pm$ 0.3 | 99.5 $\pm$ 0.1 | 95.0 $\pm$ 0.1 | 98.6 $\pm$ 0.1 |
| ProtoNet [18]        | 98.5 $\pm$ 0.2 | 99.5 $\pm$ 0.1 | 95.3 $\pm$ 0.2 | 98.7 $\pm$ 0.1 |
| *This paper*         | **99.8** $\pm$ 0.1 | **99.9** $\pm$ 0.1 | **98.3** $\pm$ 0.3 | **99.4** $\pm$ 0.2 |

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

Table A2: Performance comparison under 20-way settings on the *mini*ImageNet and *tiered*ImageNet datasets.

| | *mini*ImageNet | | *tiered*ImageNet | |
| --- | --- | --- | --- | --- |
| | 1-shot | 5-shot | 1-shot | 5-shot |
| TAML [9] | $19.73 \pm 0.65$ | $29.81 \pm 0.35$ | n/a | n/a |
| Baseline++ [3] | n/a | $38.03 \pm 0.24$ | n/a | n/a |
| *This paper* | $\mathbf{22.07} \pm 0.53$ | $\mathbf{39.98} \pm 0.27$ | $\mathbf{24.76} \pm 0.51$ | $\mathbf{41.84} \pm 0.31$ |

Table A3: Comparison (%) on *mini*ImageNet and *tiered*ImageNet using WRN-28-10 feature extractor.

| | *mini*ImageNet, 5-way | | *tiered*ImageNet, 5-way | |
| --- | --- | --- | --- | --- |
| | 1-shot | 5-shot | 1-shot | 5-shot |
| Activation to Parameter [14] | $59.60 \pm 0.41$ | $73.74 \pm 0.19$ | - | - |
| Fine-Tuning [4] | $57.73 \pm 0.62$ | $78.17 \pm 0.49$ | $66.58 \pm 0.70$ | $85.55 \pm 0.48$ |
| LEO [17] | $61.76 \pm 0.08$ | $77.59 \pm 0.12$ | $66.33 \pm 0.05$ | $81.44 \pm 0.09$ |
| *This paper* | $\mathbf{63.45} \pm \mathbf{0.39}$ | $\mathbf{78.99} \pm 0.?$ | $\mathbf{68.54} \pm \mathbf{0.61}$ | $\mathbf{86.25} \pm \mathbf{0.39}$ |

[4] G. S. Dhillon, P. Chaudhari, A. Ravichandran, and S. Soatto. A baseline for few-shot image classification. In *ICLR*, 2020.

[5] C. Finn, P. Abbeel, and S. Levine. Model-agnostic meta-learning for fast adaptation of deep networks. In *ICML*, 2017.

[6] V. Garcia and J. Bruna. Few-shot learning with graph neural networks. In *ICLR*, 2018.

[7] J. Gordon, J. Bronskill, M. Bauer, S. Nowozin, and R. E. Turner. Meta-learning probabilistic inference for prediction. In *ICLR*, 2019.

[8] A. Graves, G. Wayne, and I. Danihelka. Neural turing machines. *arXiv preprint arXiv:1410.5401*, 2014.

[9] M. A. Jamal and G.-J. Qi. Task agnostic meta-learning for few-shot learning. In *Proceedings of the IEEE Conference on Computer Vision and Pattern Recognition*, pages 11719–11727, 2019.

[10] D. P. Kingma and J. Ba. Adam: A method for stochastic optimization. *arXiv preprint arXiv:1412.6980*, 2014.

[11] G. Koch. Siamese neural networks for one-shot image recognition. In *ICML Workshop*, 2015.

[12] B. M. Lake, R. Salakhutdinov, and J. B. Tenenbaum. Human-level concept learning through probabilistic program induction. *Science*, 350(6266):1332–1338, 2015.

[13] N. Mishra, M. Rohaninejad, X. Chen, and P. Abbeel. A simple neural attentive meta-learner. In *ICLR*, 2018.

[14] S. Qiao, C. Liu, W. Shen, and A. L. Yuille. Few-shot image recognition by predicting parameters from activations. In *CVPR*, 2018.

Figure A1: The sketch of the implementation.

Table A4: Performance comparison by using various $\alpha$ on the *mini*ImageNetdataset.

|  | 1-shot | 5-shot |
|---|---|---|
| $\alpha = 0.$ | $51.28 \pm 1.70$ | $65.77 \pm 0.70$ |
| $\alpha = 0.1$ | $51.93 \pm 1.80$ | $65.71 \pm 0.90$ |
| $\alpha = 0.2$ | $52.15 \pm 1.70$ | $65.99 \pm 0.80$ |
| $\alpha = 0.3$ | $52.11 \pm 1.70$ | $65.96 \pm 0.70$ |
| $\alpha = 0.4$ | $52.10 \pm 1.90$ | $66.11 \pm 0.90$ |
| $\alpha = 0.5$ | $53.15 \pm 1.60$ | $66.93 \pm 0.70$ |
| $\alpha = 0.6$ | $53.61 \pm 1.70$ | $67.15 \pm 0.80$ |
| $\alpha = 0.7$ | $\mathbf{54.73} \pm 1.60$ | $67.37 \pm 0.80$ |
| $\alpha = 0.8$ | $53.94 \pm 1.70$ | $\mathbf{68.01} \pm 0.90$ |
| $\alpha = 0.9$ | $53.77 \pm 1.80$ | $67.53 \pm 0.90$ |
| $\alpha = 1.0$ | $53.53 \pm 1.70$ | $67.05 \pm 0.80$ |

Table A5: Performance comparison with Gumbel-softmax for memory addressing.

|  | *mini*ImageNet, 5-way | | *tiered*ImageNet, 5-way | | CIFAR-FS, 5-way | |
|---|---|---|---|---|---|---|
|  | 1-shot | 5-shot | 1-shot | 5-shot | 1-shot | 5-shot |
| Gumbel-softmax | $53.27 \pm 1.70$ | $\mathbf{68.37} \pm 0.80$ | $55.25 \pm 1.80$ | $74.57 \pm 0.80$ | $62.11 \pm 1.60$ | $77.69 \pm 0.80$ |
| **Softmax** | $\mathbf{54.73} \pm 1.60$ | $68.01 \pm 0.90$ | $\mathbf{56.88} \pm 1.71$ | $\mathbf{74.65} \pm 0.81$ | $\mathbf{63.42} \pm 1.90$ | $\mathbf{77.93} \pm 0.80$ |

[15] S. Ravi and H. Larochelle. Optimization as a model for few-shot learning. In *ICLR*, 2017.

[16] M. Ren, E. Triantafillou, S. Ravi, J. Snell, K. Swersky, J. B. Tenenbaum, H. Larochelle, and R. S. Zemel. Meta-learning for semi-supervised few-shot classification. In *ICLR*, 2018.

[17] A. A. Rusu, D. Rao, J. Sygnowski, O. Vinyals, R. Pascanu, S. Osindero, and R. Hadsell. Meta-learning with latent embedding optimization. In *ICLR*, 2019.

[18] J. Snell, K. Swersky, and R. Zemel. Prototypical networks for few-shot learning. In *NeurIPS*, 2017.

[19] O. Vinyals, C. Blundell, T. Lillicrap, K. Kavukcuoglu, and D. Wierstra. Matching networks for one shot learning. In *NeurIPS*, 2016.

[20] O. Vinyals, M. Fortunato, and N. Jaitly. Pointer networks. In *NeurIPS*. 2015.

[21] J. Weston, S. Chopra, and A. Bordes. Memory networks. In *ICLR*, 2014.

[22] S. Zagoruyko and N. Komodakis. Wide residual networks. *arXiv preprint arXiv:1605.07146*, 2016.

Table A6: The architectures of inference networks and prior network.

The inference network $q_\varphi(\cdot)$ for Omniglot, *mini*ImageNet, CIFAR-FS.

| Output size | Layers |
|---|---|
| $J \times 512$ | concatenate $\mathbf{m}^{(j)}$ and $\overline{h_\phi(\mathbf{x}_{S_n^t})}$ |
| 256 | fully connected, ELU |
| 256 | fully connected, ELU |
| 256 | linear fully connected to $\mu_z$, $\log \sigma_z^2$ |

The prior network $p_\theta(\cdot)$ for Omniglot, *mini*ImageNet, CIFAR-FS

| Output size | Layers |
|---|---|
| 256 | Input query feature |
| 256 | fully connected, ELU |
| 256 | fully connected, ELU |
| 256 | fully connected to $\mu_z$, $\log \sigma_z^2$ |

The inference network $p_\psi(\cdot)$ for Omniglot, *mini*ImageNet, CIFAR-FS.

| Output size | Layers |
|---|---|
| 256 | Input memory feature |
| 256 | fully connected, ELU |
| 256 | fully connected, ELU |
| 256 | linear fully connected to $\mu_m$, $\log \sigma_m^2$ |

Table A7: The architectures of CNN for different datasets.

The CNN architecture $h_\phi(\cdot)$ for Omniglot.

| Output size | Layers |
|---|---|
| 28×28×1 | Input images |
| 14×14×64 | *conv2d* (3×3, stride=1, SAME, RELU), dropout 0.9, *pool* (2×2, stride=2, SAME) |
| 7×7×64 | *conv2d* (3×3, stride=1, SAME, RELU), dropout 0.9, *pool* (2×2, stride=2, SAME) |
| 4×4×64 | *conv2d* (3×3, stride=1, SAME, RELU), dropout 0.9, *pool* (2×2, stride=2, SAME) |
| 2×2×64 | *conv2d* (3×3, stride=1, SAME, RELU), dropout 0.9, *pool* (2×2, stride=2, SAME) |
| 256 | flatten |

The shallow architecture $h_\phi(\cdot)$ for CIFAR-FS

| Output size | Layers |
|---|---|
| 32×32×3 | Input images |
| 16×16×64 | *conv2d* (3×3, stride=1, SAME, RELU), dropout 0.5, *pool* (2×2, stride=2, SAME) |
| 8×8×64 | *conv2d* (3×3, stride=1, SAME, RELU), dropout 0.5, *pool* (2×2, stride=2, SAME) |
| 4×4×64 | *conv2d* (3×3, stride=1, SAME, RELU), dropout 0.5, *pool* (2×2, stride=2, SAME) |
| 2×2×64 | *conv2d* (3×3, stride=1, SAME, RELU), dropout 0.5, *pool* (2×2, stride=2, SAME) |
| 256 | flatten |

The CNN architecture $h_\phi(\cdot)$ for *mini*ImageNet and *tiered*ImageNet

| Output size | Layers |
|---|---|
| 84×84×3 | Input images |
| 42×42×64 | *conv2d* (3×3, stride=1, SAME, RELU), dropout 0.5, *pool* (2×2, stride=2, SAME) |
| 21×21×64 | *conv2d* (3×3, stride=1, SAME, RELU), dropout 0.5, *pool* (2×2, stride=2, SAME) |
| 10×10×64 | *conv2d* (3×3, stride=1, SAME, RELU), dropout 0.5, *pool* (2×2, stride=2, SAME) |
| 5×5×64 | *conv2d* (3×3, stride=1, SAME, RELU), dropout 0.5, *pool* (2×2, stride=2, SAME) |
| 2×2×64 | *conv2d* (3×3, stride=1, SAME, RELU), dropout 0.5, *pool* (2×2, stride=2, SAME) |
| 256 | flatten |