[Reviews · NeurIPS 2020]

Review 1

Summary and Contributions: The authors propose to construct a semantic memory module based on Prototypical Networks in order to boost the performance of few-shot recognition. Specifically, they model the memory recall part as a variational inference framework, such that the latent memory and class prototypes are stochastic. The experimental results suggest that the proposed method improves on the base Prototypical Networks on various standard benchmark datasets for few-shot classification.

Strengths: The main strength of this paper is that it provides a novel perspective on few-shot classification that we need a semantic memory module to better structure the previous experiences. In my opinion, although not discussed in the paper, this intuition can be useful for the combination of continual learning and meta-learning, as an external memory module could help avoiding catastrophic forgetting by structuring and well preserving the past experiences. And the experimental results show that the proposed method is competitive with some of the recent few-shot recognition methods on various benchmark datasets and architectures.

Weaknesses: The main weakness of this paper is correctness and clarity. The detailed discussions are provided below. In my opinion, the correctness part is below the NeurIPS acceptance threshold, and it should be improved a lot. The second but less significant limitation is that, from Table 1 and 2, the major performance improvements come from "Variational prototype network", which is quite orthogonal to the main argument of this paper that making use of semantic memory is important for few-shot recognition.

Correctness: As mentioned above, I am a bit skeptical about the technical correctness for the variational inference framework. Specifically, - I think the latent z in Eq.(2) does not properly represent the class prototypes as z is conditioned on each individual x, not a entire class set (But on the other hand, Figure 1 shows that the latent z is conditioned on each of the class sets, and I'm confused which one is right). - Also in Eq.(2), it seems that the prototype variable z is conditioned on the query instances, which is weird because prototype should represent each support set. (In my opinion, in order to properly represent the variational inference framework for meta-learning, it would be better to use the framework of Neural Processes [1].) - In Eq.(3), I don't understand how the approximate posterior q(z|S) can have dependency on S, because according to the generative process defined by Eq.(2), the true posterior p(z|x,y) does not have the dependency on the entire class set S except for each individual point (x,y). - In wonder whether the intermediate variable m in Eq.(4) is included in the generative process as well, or not. If it is not included, then the inference of m should be based on semi-implicit variational inference [2,3] as the intermediate stochastic variable m is only for the approximate posterior. However, such a discussion has not been discussed in the paper and the ELBO expression Eq.(13) seems not to represent the SIVI procedure as well. Same for Eq.(5). - In Eq.(6), I don't understand why the first summation term appears. Also, I guess p(m|M_a, S) corresponds to q(m|M,S) in Eq.(5)? - I cannot find the definition of q(z|S,m)? - In Eq.(13), I don't understand how the second KL term appears. = Reference = [1] Garnelo et al., Neural Processes, arxiv 2018. [2] Yin et al., Semi-implicit Variational Inference, ICML 2018. [3] Molchanov et al., Doubly Semi-implicit Variational Inference, AISTATS 2019.

Clarity: I think the clarity of this paper is below the acceptance threshold as I guess many parts in the approach section are incorrect. - Also, in the motivation side, it would be good to add more discussions and intuitions about why making use of semantic memory module should produce better results than the traditional Prototypical Networks. - I guess that the graphical model in Figure 1 represents the computational graph, which consists of both generative process and posterior inference part. It would be better to distinguish between the two (e.g. bold vs. dotted line) for better readability. - I would be good to discuss a little bit in more detail about the form of each diagonal gaussian distribution.

Relation to Prior Work: There seems no significant problem with this part.

Reproducibility: Yes

Additional Feedback: I have some more questions. - Do the query instances also refer to the memory module? In other words, are the two inference pipelines for support and query set the same? - What is the use of batch normalization? Is it transductive via batch normalization or not? - Figure 3 suggests that the performance can be further improved if the memory size increases further. Why did you stop at 64? Is it because of some computational issue? - In Figure 2, the shape of the distributions are all unimodal, although the hierarchical posterior distribution defined in earlier sections implies that it would be multimodal. Could you explain about this? Overall, I think the submission is below the acceptance threshold. I suggest the authors to revise the correctness of the variational inference part.


Review 2

Summary and Contributions: The paper proposes a latent variable memory that enables consolidation of concepts in a semantically similar manner. The memory improves the downstream task of few shot classification in a variety of standard few shot learning benchmarks. Overall the paper is presented in a clean Bayesian manner, incorporating a variational lower bound that captures the objective in a meaningful probabilistic manner.

Strengths: 1. The paper is well written and easy to understand. 2. The proposed method is SOTA on all tested tasks. 3. The use of graph based memory updates are a novel contribution as far as I can tell. 4. The proposed Bayesian framework is elegant. 5. The ablation against a non-distributional memory are appreciated.

Weaknesses: My main concern with this work is the lack of comparison to other (highly) similar memory models such as the Kanerva Machine (KM) [2] & Dynamic Kanerva Machine (DKM) [3]. As far as I can tell there are no other memory models such as NTM / DNC / KM / DKM used in the baselines. The authors of [2, 3] show a drastic performance differential against non-memory models due to their ability to fuse information across an episode of samples (in the case studied here it would be fusing information across episodes). A fair evaluation of a memory model should include other memory models. ** I will be reading the author responses and will change my review given that the authors address / clarify the points from this review ** [2] Wu, Yan, et al. "The Kanerva Machine: A Generative Distributed Memory." International Conference on Learning Representations. 2018. [3] Wu, Yan, et al. "Learning attractor dynamics for generative memory." Advances in Neural Information Processing Systems. 2018.

Correctness: 1. Is there a performance difference when approximating the addressing vector as a categorical (through VIMCO or the Gumbel-softmax estimator) instead of the softmax approximation? 2. I tried to run your code using a fresh anaconda environment with tensorflow-1.15 and the misc missing dependencies, but got a variable-reuse error. I was thus not able to directly validate the code. While code & the provided bash scripts are greatly appreciated, consider a docker container for the future to ensure a smooth workflow.

Clarity: The paper is well written and easy to understand. The biological parallels are interesting and aid in pushing the envelope of why we should use memory models, but some of the elucidations are a bit of a stretch. A few minor points should be addressed to make the paper cleaner: 1. How important is the EMA on the memory? Does setting the decay at 0.9999 vs 0.8 drastically alter performance? This can be added to the appendix, but is an important part of the memory updates and should be quantified. 2. While the ablation on variational memory / raw memory is informative, it is not clear if the same applies to the conditioned variable z. Does z need to be present? And if so, does it need to be stochastic?

Relation to Prior Work: 1. The paper is heavily related to [1] in its addressing mechanism (albeit in a smoother manner that linearly combines memory rows) and [2, 3] in its usage of latent variable memory, however [1] is only described in one sentence. [3] is not even referenced as a relevant work and [2] is incorrectly stated to have collapsing keys (see point 3 below). 2. While [2, 3] use a latent variable matrix gaussian, the addressing mechanism in practice becomes a linear combination of the read vector and the rows of the matrix gaussian mean (mentioned in [2]). While this paper presents SOTA in its target domain of few-shot learning there are no other memory models used as baselines. The authors of [2,3] showed that the Kanerva Machine drastically outperformed non-memory baselines and demonstrate its improvement over the DNC. 3. The KM and DKM models do not collapse to single slots as far as I'm aware since the keys have their own learned amortized approximate posteriors. Line 51 is correct for DNC, but not for KM/DKM. [1] Bornschein, Jörg, et al. "Variational memory addressing in generative models." Advances in Neural Information Processing Systems. 2017. [2] Wu, Yan, et al. "The Kanerva Machine: A Generative Distributed Memory." International Conference on Learning Representations. 2018. [3] Wu, Yan, et al. "Learning attractor dynamics for generative memory." Advances in Neural Information Processing Systems. 2018.

Reproducibility: No

Additional Feedback: **Post Rebuttal Feedback** : I'm content with the author's responses and the addition of MANN, however I would recommend adding a comparison to DKM/KM as they are highly related. Thanks for the info regarding the EMA and Gumbel-Softmax estimator.


Review 3

Summary and Contributions: The paper introduces a probabilistic modeling framework of the prototypical networks using variational inference. The model is augmented by an external memory, called variational semantic memory. The experiment results show the benefit of the proposed model in comparison to the previous models on mini/tieredImageNet and CIFAR-FS

Strengths: Overall clearly written although some improvements of the description structure would make reading easier. Probabilistic modeling of protypical representation is interesting. And the performance gain with the external memory is impressive. The experiment result is good overall.

Weaknesses: - Memory 'M' is discussed from the 'memory recall and inference' but what it actually means is detailed only later in the 'Memory update and consolidation' section. It would be better to introduce it earlier. - For without memory version, how q(z|S) is implemented is not clear. - In the model in Eqn(2), why not condition the model (prior and likelihood) on S? Similarly, why not condition the variational posterior q(z|S) on (x_i, y_i) as well? - The memory is basically constructed using the class label because for a new class the representation is just appended to the memory. Only the updated of the semantic representation is learned while learning the sematic (clustering) is done by provided labels. In this sense, I'm afraid to say that the term "semantic" is a bit overly used. - The memory mechanism seems not that novel because its base is on variational memory addressing and with some update like graph attention. - In Eq 8, the distance function g(M_a, S), is not clearly described. M_a is a vector and S is a set. How do you define a dot product. - In Eqn (13), "L = argmin" should be "argmin L" - Lack of some analysis on how modeling uncertainty (without semantic memory) helps. Table 1. shows the overall performance improvements but it's not clear how uncertainty contribute to this. - Some ablation study, such as without using the graph attention for memory update (which I think become similar to VMA reading) would be informative. - Discussion and reference to some Bayesian approaches to Meta-Learning is missing, e.g. Bayesian MAML (J Yoon et. al., ‎2018) and Amortized Bayesian Meta-Learning (S Ravi - ‎2018) - Discussion about "Gaussian Prototypical Networks for Few-Shot Learning on Omniglot" is missing

Correctness: The model and empirical method is correct.

Clarity: Clear overall. The structure of the description can be improved though. (See weaknesses)

Relation to Prior Work: Discussion on some important related works are missing (see Weaknesses)

Reproducibility: Yes

Additional Feedback:


Review 4

Summary and Contributions: This submission tackles to mitigate the data scarcity in the few-shot learning problem. This work proposes a variational memory-based prototype network that uses probabilistic prototype representation rather than deterministic one in ProtoNet [59]. The proposed method can be viewed as a non-trivial extension of ProtoNet. This idea is reasonable because, in the scarce data regime, uncertainty would be non-negligible, and also rare class data can be effectively dealt with long term memory. The proposed algorithm derivation is clear and intuitive. Also, they demonstrate the effectiveness of the proposed method successfully. It would have been complete if the authors follow the standard benchmark protocol, e.g., 20-way experiments are missed. =========== updated after rebuttal =========== This reviewer has read all the other review comments and the rebuttal, and found that all the questions and concerns are well dealt with by the authors' rebuttal. I still believe that this work has its potential value sufficient to report in the community by showing that uncertainty modeling and latent parameterization by semantic memory effectively improve ProtoNet in the few-shot recognition. Thus, this reviewer votes for acceptance of this work, and increase the score. While this work is interesting, as pointed by other reviewers, the paper description, structure, and details should be improved because the missing details or ambiguous descriptions raise most of the concerns. Thus, the authors should reflect all the comments and put efforts to resolve the concerns in the camera-ready. Also, it would have been stronger if the analysis is more through to understand where the performance improvement specifically comes from.

Strengths: - Hierarchical and probabilistic design of ProtoNet - Noticeable performance improvement and favorable performance against the state-of-the-art methods. - The technical design is well-motivated

Weaknesses: - 20-shot experiments are completely missing, which have been evaluated from the prior arts. - Missing reference (see below)

Correctness: - The algorithm derivation is well developed.

Clarity: - The paper is written very well.

Relation to Prior Work: - Properly discussed the innovation compared to ProtoNet well. Missing reference: Although this submission is more general and extended work, the following work seems very closely related. [Variational Prototyping-Encoder: One-Shot Learning with Prototypical Images, CVPR2019]

Reproducibility: Yes

Additional Feedback: - As a small suggestion, since the proposed method is good in uncertain data regimes, it may demonstrate more strong advantages of the proposed method if a zero-shot experiment is added as done in ProtoNet. - The authors only show the limited memory size cases. Since the performance trend does not show the saturation yet, it would be informative to show more experiments up to a saturation point. Does it have any computational burden problem? - The proposed method requires two MC samplings. How much do those samplings affect the training time and stability of the training?

[Author Response · NeurIPS 2020]

*We thank all four reviewers for their candid feedback and sharp comments.*

**Correctness and Clarity (R1, R3)**    We regret our method description has confused R1 and R3, but we believe it
is still mendable. Instead of directly inferring $\mathbf{z}$ from each individual query sample $\mathbf{x}$, we introduce the variational
posterior $q(\mathbf{z}|S)$ in Eq. (2) and (3) to make it dependent on support set $S$ and suitable for the few-shot setting. $S$
infers $\mathbf{z}$ and predicts class $\mathbf{y}$ of the sample $\mathbf{x}$. $p(\mathbf{z}|\mathbf{x})$ is the conditional prior, serving only as a regularizer through
the KL-term in the ELBO. R3 wonders why we do not condition the entire model on $S$ and the variational posterior
on $(\mathbf{x}_i, \mathbf{y}_i)$. Conditioning $q(\mathbf{z}|S)$ only on $S$ fits the prototypical few-shot recognition; likelihood only depends on $\mathbf{x}$
and $\mathbf{z}$, no longer on $S$. $q(\mathbf{z}|S)$ is not conditioned on $(\mathbf{x}_i, \mathbf{y}_i)$ because the label $\mathbf{y}_i$ of $\mathbf{x}_i$ is unknown. To R1: latent
memory $\mathbf{m}$ in Eq. (4) is indeed only for the approximate posterior, not for generative process; only support (not
query) instances refer to memory (see also Fig. 1). Both $q(\mathbf{z}|\mathbf{m}, S)$; and $p(\mathbf{m}|S)$ are explicit with analytic PDFs, so
we are not based on SIVI, the same for Eq. (5). The first summation in Eq. (6) is derived from Eq. (5) as follows:
$\tilde{q}(\mathbf{z}|M, S) = \int q(\mathbf{z}|S, \mathbf{m}, M)q(\mathbf{m}|M, S)d\mathbf{m} = \int q(\mathbf{z}|S, \mathbf{m})q(\mathbf{m}|M, S)d\mathbf{m}$. In response to R1 and R3, all conditional
distributions are implemented as uni-modal diagonal Gaussian: for $q(\mathbf{z}|S)$ we take the representation of the 1-shot
sample or the average of 5-shot samples in $S$ as input and return $\mu$ and $\sigma$ of $\mathbf{z}$. Finally, the second KL-term in Eq. (13)
is between the variational posterior and the conditional prior, derived from hierarchical variational inference, ensuring
that the inferred latent memory is indeed relevant to the current few-shot recognition task. We will expand the method
description accordingly. Thank you.

**Related Works (R1, R2, R3, R4)**    We apologize to R2 for missing Wu et al. NeurIPS 2018. To R2 and R3, the major
difference with Wu et al. (and Bornschein et al. [8] and Wu et al. [72]) is that we treat retrieved memory content – not
the addressing vector – as the stochastic variable. This enables retrieved content to be better adapted to the current
few-shot task. We meant that KM [72] *avoids* collapsing memory reading and writing into single memory slots, we will
rephrase L50-51. We infer distributions of prototypes directly from the support sets (leveraging semantic memory), not
by reconstruction as Kim et al. CVPR 2018. We will also include the other suggested references and elaborate on our
relation to SIVI, Neural Process, Bayesian meta-learning and Gaussian prototypical network for few-shot learning and
variational prototyping-encoder for one-shot learning.

**Comparisons and Ablations (R1, R2, R3, R4)**    By request of R2 we add
a comparison with MANN [55] (an instance of NTM for meta learning) in
Table I. Indeed KM [72] and DKM (Wu et al. NeurIPS 2018) could also
be explored for few-shot learning; we will implement and add them to our
comparison as well. In response to R2's question on alternatives for the
softmax approximation, we have implemented Gumbel-softmax. Results
on *mini*ImageNet are slightly worse on 1-shot (-0.46%) and slightly better

**Table I: Comparison with MANN (R2)**

|  | *mini*ImageNet | | *Omniglot* | |
|---|---|---|---|---|
|  | 1-shot | 5-shot | 1-shot | 5-shot |
| MANN | 41.38 | 61.73 | 93.5 | 97.6 |
| **Ours** | **54.73** | **68.01** | **99.8** | **99.9** |

on 5-shot (+0.36%). Note Gumbel-softmax needs a predefined memory size, where our proposal can deal with
dynamic memories with growing size. EMA controls how much old information will be forgotten during memory
update. We choose the decay by cross validation; when varying decay from 0.8 to 0.9999 performance drops slightly
(1-shot: 1.2%, 5-shot: 0.5%). We will add it to the appendix. We apply memory to conditioned $\mathbf{z}$, as already detailed
in the Appendix. While $\mathbf{z}$ does not need to be stochastic, Table 1 reveals it is much better than a deterministic one.
R4 asks for 20-way experiments. Besides the 20-way results on Omniglot in Table A1 of the Appendix, we report 20-
way results and comparison with state-of-the-art (Jamal et al. CVPR 2019 and Chen et al. ICLR 2018) on *mini*ImageNet
in Table II. We did not find prior 20-way results on *tiered*ImageNet. The requested ablation by R3 on graph attention is
provided in Table A3 of the Appendix. Note our graph attention is for updating the memory, not for reading as in VMA.
To R3, the improvement over ProtoNet is mainly due to the uncertainty
modeling by our variational prototype net, as the major difference is modeling
prototypes as probabilistic distributions rather than deterministic vectors. The
probabilistic prototypes innately model uncertainty by producing distributions
over predictions, which offers better confidence calibration than deterministic
ones. Table 1 shows the performance benefit due to the ability of modeling
uncertainty, while Fig. 2 provides the intuitive illustration. We will add more

**Table II: 20-way results (R4)**

|  | *mini*ImageNet | | *tiered*ImageNet | |
|---|---|---|---|---|
|  | 1-shot | 5-shot | 1-shot | 5-shot |
| SOTA | 19.73 | 38.03 | n/a | n/a |
| **Ours** | **22.07** | **39.98** | **24.76** | **41.84** |

analysis on modeling uncertainty. To R1, R4, the maximum memory size is the total number of classes. We report up
to the 64 classes in *mini*ImageNet. Performance could increase with more categories. Our memory is compact and
has no computational burden problem. To R4, we choose sampling rates by cross validation. Indeed, larger rates can
increase training time while it would make training more stable. We will add this ablation in the Appendix. Thanks for
the clever suggestions to try our model on zero-shot and continual learning tasks, which we will definitely explore in
future work. In response to R1, transductive BN is used for fair comparison with previous works. We will release a
docker container with all source code.

*We thank all reviewers for their feedback and the opportunity for reconsideration.*

[Meta-Review · NeurIPS 2020]

This paper adds uncertainty modelling and an external memory to prototypical networks and achieves good performance gains as a result. The paper is above the bar for acceptance. There were a number of extra details, clarifications, and related work that was brought up in the reviews and rebuttal that should be incorporated into the final camera ready version. The authors should also compare to the DKM model and try to quantify the benefit of uncertainty modelling, as promised in the rebuttal.